# How Is Occam's Razor Realized in Symbolic Regression?: An Adaptive LLM-Enhanced Genetic Programming Approach for Efficient, Versatile, and Interpretable Representation Discovery through Simplification and Evolution

## Abstract

Symbolic regression aims to discover mathematical expressions that capture underlying data relationships, but genetic programming (GP) approaches commonly encounter bloat, premature convergence, and inadequate expression simplification mechanisms. We propose ALEGP (Adaptive LLM-Enhanced Genetic Programming), a framework that strategically integrates large language models (LLMs) with evolutionary computation to address these interconnected challenges. ALEGP incorporates three key components: (i) a multi-island evolutionary architecture employing specialized subpopulations with distinct optimization objectives to maintain population diversity, (ii) a context-aware intervention scheduler that triggers LLM assistance based on real-time evolutionary indicators including fitness stagnation, diversity loss, and expression bloat, and (iii) an island-specific integration protocol that reincorporates LLM-refined expressions while preserving beneficial evolutionary dynamics. This design enables targeted simplification of complex expressions, improved generalization performance, and reduced computational overhead through adaptive LLM utilization. Experiments on eight synthetic benchmark functions and five real-world regression datasets demonstrate that ALEGP achieves superior accuracy and interpretability while requiring 50–60% fewer LLM interventions than fixed-schedule strategies. Ablation studies validate the necessity of both adaptive scheduling and multi-island design for robust performance. These results establish ALEGP as an effective framework for resource-efficient symbolic regression, demonstrating principled integration of evolutionary algorithms with large language models. Code is provided as supplementary material.

## 1 Introduction

Symbolic regression (SR) discovers mathematical expressions that capture relationships between input and output variables Zhong et al. (2025). Unlike parametric regression, SR constructs formulas using finite primitive sets without assuming predetermined functional forms Huang et al. (2024). This flexibility enables widespread adoption across industrial data analysis Quade et al. (2016), knowledge discovery Luna et al. (2014), data mining Krömer et al. (2013), and time series prediction Dabhi & Chaudhary (2016).

Genetic programming (GP) Koza (1994) has emerged as the dominant SR approach due to its effectiveness in evolving symbolic trees into interpretable mathematical formulas. GP models Darwinian evolution by evolving candidate program populations encoded as syntax trees, using mutation and crossover operators to search the solution space. However, GP faces several critical limitations in SR applications. First, GP suffers from bloating—continuous tree growth without fitness improvement—resulting in unnecessarily complex expressions that reduce interpretability and increase overfitting risk dal Piccol Sotto & de Melo (2016); Rimas et al. (2023). Second, premature convergence and local optima stagnation limit solution space exploration. Third, GP lacks

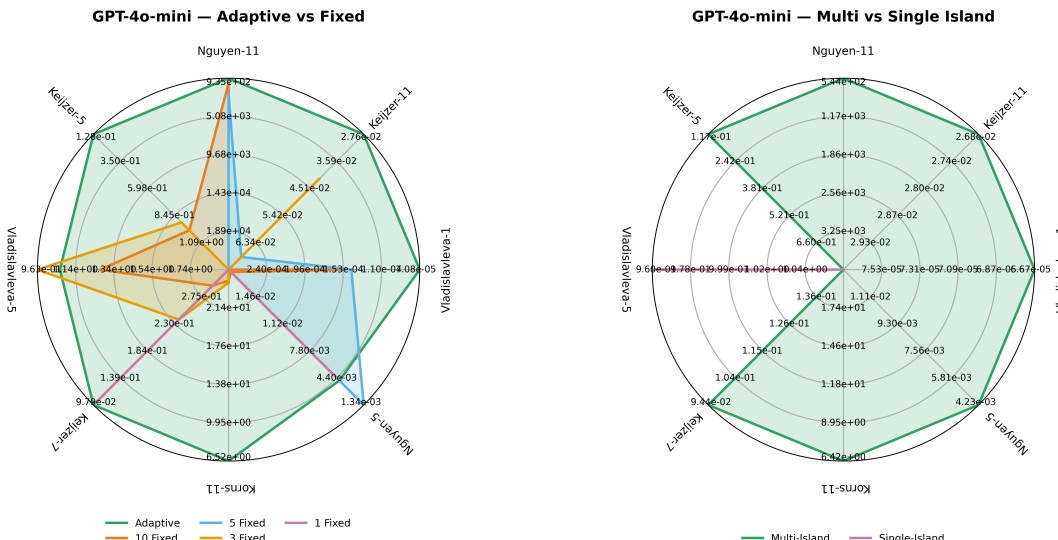

Figure 1: Radar plots illustrating normalized performance across representative symbolic regression benchmarks. (Left) Adaptive versus fixed intervention strategies. (Right) Multi-island versus single-island architectures. Larger enclosed areas indicate better performance (lower MSE, higher correlation). The plots highlight two consistent trends: adaptive scheduling provides more stable gains, and multi-island frameworks achieve broader improvements.

sophisticated mechanisms to transform complex expressions into concise, generalizable forms Haeri et al. (2017).

Researchers have addressed these limitations through various approaches, including multi-form GP frameworks Zhong et al. (2025), domain-specific genetic operators Huang et al. (2024), and adaptive strategies Chen et al. (2019b;a); Haeri et al. (2017). Recently, several studies have focused on integrating artificial intelligence (AI) methods into GP, such as neural-guided approaches Mundhenk et al. (2021), denoising autoencoder Wittenberg & Rothlauf (2022; 2023), transformers Jha et al., and gradient-based techniques Pietropolli et al. (2023), weighted K-nearest neighbor Al-Helali et al. (2020) and relevance vector machine (RVM) Iba et al. (2018). However, these approaches typically address individual limitations rather than tackling all challenges simultaneously. For instance, gradient-based methods [Pietropolli et al. (2023) effectively reduce bloat but fail to address premature convergence, while neural-guided approaches Mundhenk et al. (2021), improve exploration but may add substantial computational overhead without providing simplification mechanisms.

To this end, we propose an *Adaptive LLM-Enhanced Genetic Programming (ALEGP)*, a novel and innovative approach that leverages LLMs within the GP framework to simultaneously address multiple limitations of traditional GP for symbolic regression. The proposed ALEGP approach capitalizes on the complementary strengths of both paradigms: GP provides strong search capabilities from evolutionary processes but suffers from bloat and local optima issues and LLMs offer sophisticated mathematical reasoning abilities that can simplify complex expressions and help escape evolutionary stagnation points. Our key technical contributions and breakthroughs in this work include the followings:

- A multi-island evolutionary framework with LLM-driven expression refinement that maintains diverse subpopulations using specialized evolutionary strategies and transforms promising candidates into more generalizable and interpretable forms through LLM mathematical reasoning capabilities.

- An adaptive intervention scheduler that monitors evolutionary dynamics to optimize LLM assistance timing, and an expression integration protocol that incorporates refined expressions back into the evolutionary process using island-specific strategies with continuous adaptive learning.

- Comprehensive experimental validation on synthetic benchmark functions and real-world datasets demonstrates significant improvements in accuracy, interpretability, and expression simplicity compared to baseline algorithms.

Figure 1 presents radar plots comparing (i) adaptive versus fixed intervention strategies and (ii) multi-island versus single-island architectures on representative symbolic regression benchmarks. Each axis represents a normalized performance metric, where larger enclosed regions indicate superior performance across multiple measures. The plots demonstrate two key findings: adaptive scheduling consistently outperforms fixed schedules with improved stability, while multi-island frameworks achieve broader performance gains relative to single-island baselines. These preliminary results, detailed further in Section 4 and Appendix B, demonstrate ALEGP's effectiveness in addressing fundamental GP limitations including bloat, premature convergence, and limited exploration diversity.

## 2 RELATED WORKS

**Symbolic Regression.** Given a dataset $\mathcal{D} = \{(\mathbf{x}_i, y_i)\}_{i=1}^N$ where $\mathbf{x}_i \in \mathbb{R}^d$ and $y_i \in \mathbb{R}$, symbolic regression seeks to discover a mathematical expression $f : \mathbb{R}^d \to \mathbb{R}$ itself that minimizes the error metric (typically mean squared error):

$$\mathcal{L}(f) = \frac{1}{N} \sum_{i=1}^N (f(\mathbf{x}_i) - y_i)^2$$

Symbolic regression does not assume a predetermined functional form, unlike parametric regression approaches. Instead, it searches for expressions composed of basic mathematical operations, variables, and constants. The challenge lies in finding expressions that balance accuracy with simplicity and generalizability.

**GP for Symbolic Regression.** Researchers have developed diverse approaches to address the fundamental limitations of GP in symbolic regression. Few recent advances include different efforts at lifetime learning with hill climbing Azad & Ryan (2014), statistical metric-based search space structuring Haeri et al. (2017), angle-driven semantic GP Chen et al. (2019a), and risk-minimization for balancing complexity Chen et al. (2019b). Additional innovations include multi factorial evolution Zhong et al. (2020), problem decomposition Liu et al. (2021), and transformed semantics for exploration-exploitation balance Chen et al. (2021). Federated methods address privacy concerns Dong et al. (2023), while formal constraints Błądek & Krawiec (2023) and knowledge transfer Al-Helali et al. (2023) improve quality of expression. Notable recent advances in representation include semantic linear GP Huang et al. (2024) and multiform frameworks with dynamic resource allocation Zhong et al. (2025), along with feature relevance assessment Chen et al. and semantically-aware operators Pawlak & Krawiec (2018).

**Integrating AI Methods with GP for Symbolic Regression.** The integration of AI approaches into GP for symbolic regression has substantially enhanced capabilities across multiple dimensions. Specifically, neural-guided GP approaches utilize recurrent networks to seed populations and retain their evolutionary exploration Mundhenk et al. (2021), while transformer-based approaches enhance robust pattern recognition and expression refinement Jha et al.. Further, GP trees have been converted into functions viable for gradient optimization via parameter embedding Pietropolli et al. (2023) in addition to autoencoder architectures using basis edit distances that respect syntactic validity Wittenberg & Rothlauf (2022; 2023). Moreover, specialized approaches address incomplete data through feature selection with K-nearest neighbors Al-Helali et al. (2020) and develop Bayesian sparse kernel methods for interpretability Iba et al. (2018).

**Our Research Motivations.** Despite significant advancements in symbolic regression, current approaches usually resolve GP drawbacks individually rather than comprehensively tackling the interconnected challenges of bloat reduction, convergence acceleration, and expression simplification. Integrating LLMs with GP yields a promising solution to address these challenges. However, the currently proposed integration methods concentrate on population initialization, refinement, or guidance of evolutionary trajectories, frequently using fixed-interval interventions or prescriptive transformation strategies that do not accommodate the evolving nature of evolution. This rigid

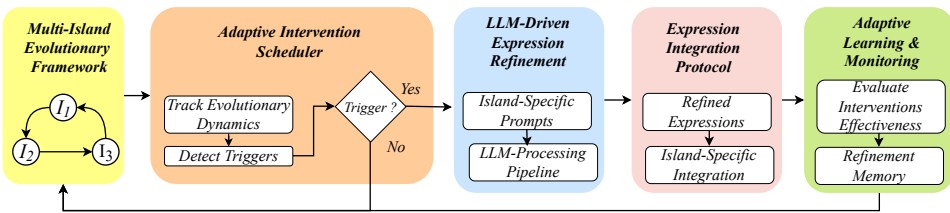

Figure 2: General Framework of the Proposed ALEGP

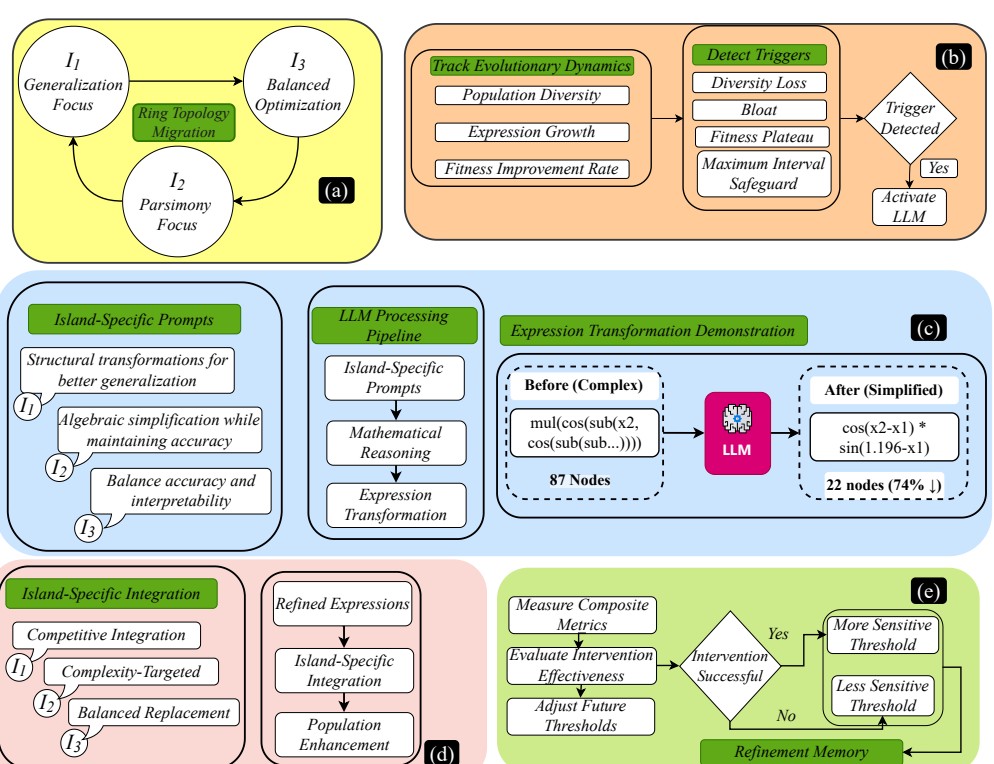

Figure 3: Detailed Components of the Proposed ALEGP Framework. (a) Multi-Island Evolutionary Framework (b) Adaptive Intervention Scheduler (c) LLM-Driven Expression Refinement Mechanism (d) Expression Integration Protocol (e) Adaptive Learning & Success Monitoring

integration paradigm leads to computational inefficiency and suboptimal performance, except if interventions happen at evolutionary hotspots. Our Adaptive LLM-Enhanced Genetic Programming (ALEGP) overcomes these deficiencies via context-aware interventions, a multi-island architecture that preserves evolutionary diversity, and dynamic expression refinement.

## 3 OUR APPROACH: ALEGP

This section presents the theoretical framework for Adaptive LLM-Enhanced Genetic Programming (ALEGP), which integrates LLMs into GP for symbolic regression to address bloat, premature convergence, and local optima. Unlike fixed-interval LLM interventions Mundhenk et al. (2021); Petersen et al. (2019), ALEGP dynamically identifies optimal intervention conditions, maximizing impact while optimizing computational resources. The LLM component operates without ground truth access to target functions. The LLM receives only evolved expressions, evaluation metrics, and

mathematical context, ensuring improvements derive from general mathematical reasoning rather than memorizing benchmark functions.

As illustrated in Figure 2, ALEGP comprises five coordinated components that enhance symbolic regression. The multi-island evolutionary framework maintains diverse subpopulations while the adaptive intervention scheduler monitors evolutionary state. When the scheduler detects critical conditions—fitness stagnation, diversity loss, or expression bloat—it triggers LLM-driven expression refinement for targeted mathematical transformations. The expression integration protocol then incorporates refined solutions using specialized replacement strategies. The adaptive learning and monitoring component provides continuous feedback to optimize future interventions. This architecture addresses fundamental GP limitations while optimizing computational resources through selective LLM intervention.

## 3.1 MULTI-ISLAND EVOLUTIONARY FRAMEWORK

Figure 2(a) shows the multi-island architecture maintaining three specialized subpopulations connected through ring topology migration. A critical challenge in GP-based symbolic regression is maintaining population diversity while ensuring effective search direction. ALEGP addresses this through a multi-island model where subpopulations evolve in parallel with periodic migration, inspired by allopatric speciation in biological evolution Tomassini (2005). This model supports diverse evolutionary trajectories while maintaining focused search in promising regions. Each island maintains a population of candidate expressions represented as parse trees, with operations (arithmetic, trigonometric, etc.) as internal nodes and variables or constants as leaves. The critical innovation lies in each island's distinct evolutionary parameters and specialization focus:

1. **Island $I_1$**: Emphasizes *generalization capability* with higher mutation rates ($\mu_1 = 0.2$) and tournament selection with strong selective pressure
2. **Island $I_2$**: Focuses on *expression parsimony* with moderate mutation rates ($\mu_2 = 0.1$) and higher crossover probability ($\chi_2 = 0.7$)
3. **Island $I_3$**: Seeks *balanced performance* with intermediate parameter values ($\mu_3 = 0.15$, $\chi_3 = 0.65$)

Ring topology migration facilitates genetic exchange while preserving unique evolutionary characteristics. This heterogeneity enables diverse evolutionary trajectories while reducing system-wide convergence to local optima.

## 3.2 ADAPTIVE INTERVENTION SCHEDULER

Figure 3(b) details the adaptive intervention mechanism that monitors evolutionary dynamics to optimize LLM refinement timing. A key challenge in hybrid evolutionary-LLM systems is determining optimal LLM invocation timing, as fixed-interval approaches waste computationally expensive calls while manual tuning requires domain expertise. The adaptive intervention scheduler addresses this by dynamically monitoring evolutionary state to identify optimal refinement opportunities.

The scheduler tracks three key evolutionary indicators:

1. **Population Diversity:** Quantifies diversity as the proportion of structurally unique expressions across all islands. Rapid decline suggests premature convergence, where LLM intervention can introduce beneficial variation.
2. **Expression Growth:** Monitors average expression size change between generations. When expressions grow increasingly complex without corresponding fitness improvements, this indicates bloat—a common GP issue where LLM simplification becomes valuable.
3. **Fitness Improvement Rate:** Tracks relative improvement in best fitness between generations. Consistently low improvement rates across multiple generations suggest evolutionary stagnation, where LLM-driven transformations might help escape local optima.

Based on these indicators, the scheduler employs four trigger mechanisms:

1. **Diversity Loss**: Population diversity declines sharply.
2. **Bloat**: Complexity grows without fitness gain.
3. **Plateau**: Fitness stagnates across generations.

4. **Interval Safeguard**: Enforces periodic LLM calls.

The scheduler adaptively adjusts decision thresholds based on intervention outcomes. After each refinement, the system evaluates success using a composite metric balancing fitness improvement and complexity reduction. Successful triggers become more sensitive while ineffective ones become less sensitive, tailoring intervention strategies to problem-specific characteristics and optimizing computational resources.

### 3.3 LLM-DRIVEN EXPRESSION REFINEMENT

Figure 3(c) illustrates the LLM processing pipeline that transforms complex expressions into more interpretable forms. Traditional GP algorithms struggle to transform complex expressions into elegant, generalizable forms. The LLM-driven refinement mechanism addresses this limitation by utilizing mathematical reasoning capabilities for intelligent symbolic manipulation.

- **Island 1**: Structural transformations for generalization (may trade minor accuracy).
- **Island 2**: Algebraic simplification and complexity reduction with accuracy preserved.
- **Island 3**: Balances accuracy and interpretability.

The LLM operates without ground truth access, ensuring improvements derive from mathematical reasoning rather than memorization. The prompts include exemplars of successful transformations from previous generations, establishing an adaptive learning loop that enhances refinement capability. As shown in Figure 3(c), a complex 87-node expression `mul(cos(sub(x2, cos(sub(sub...))))))` transforms into a simplified 22-node equivalent $\cos(x_2 - x_1) \cdot \sin(1.196 - x_1)$, achieving 74% complexity reduction.

### 3.4 EXPRESSION INTEGRATION PROTOCOL

Figure 3(d) shows how refined expressions are strategically incorporated into island populations. After LLM refinement, transformed expressions must be effectively integrated into existing populations. This integration presents a challenge: introducing too many refined expressions risks disrupting beneficial evolutionary patterns, while introducing too few might limit their impact. This challenge is addressed through island-specific integration strategies that align with each island's evolutionary objectives:

1. Island 1: Uses **Competitive Integration** where refined expressions compete with existing individuals through tournament selection, replacing those that perform poorly. This maintains selection pressure toward accurate models.

2. Island 2: Employs **Complexity-Targeted Replacement** where refined expressions replace the most complex expressions with comparable or better fitness. This directly addresses bloat while preserving accuracy.

3. Island 3: Implements **Balanced Replacement** using a hybrid approach that considers fitness and complexity when selecting replacement expressions. This supports the island's focus on balancing multiple objectives.

This scheme maintains evolutionary dynamics while effectively incorporating LLM insights. Each island retains its specialized role while leveraging targeted LLM improvements. Each island's replacement rate $\rho_i$ is dynamically adjusted based on historical intervention efficacy and current evolutionary stage, enabling more aggressive integration during early exploration and more conservative approaches as populations converge.

### 3.5 ADAPTIVE LEARNING AND MONITORING

Figure 3(e) illustrates the continuous evaluation and threshold adjustment mechanism that enables adaptive learning throughout the evolutionary process. Following each refinement event, the system evaluates intervention success using composite metrics that balance fitness improvement against complexity reduction. This evaluation creates a dynamic feedback loop where:

1. **Successful interventions** increase trigger sensitivity for similar evolutionary conditions, enabling more responsive future activations

2. **Less beneficial interventions** decrease sensitivity to avoid unnecessary computational overhead

3. **Refinement memory** accumulates successful transformation patterns, building a knowledge base that enhances future mathematical reasoning capabilities

This adaptive mechanism tailors intervention strategy to problem-specific characteristics, efficiently allocating computational resources where they yield maximum performance gains.

The complete ALEGP framework operates as a unified system where specialized islands generate diverse candidate solutions through distinct evolutionary strategies, the adaptive scheduler identifies optimal intervention timing based on real-time evolutionary dynamics, LLM refinement performs intelligent expression transformation using mathematical reasoning, and strategic integration protocols preserve beneficial evolutionary patterns while incorporating refined solutions. This synergistic architecture systematically addresses fundamental GP limitations—bloat, premature convergence, and inadequate simplification—while establishing a novel paradigm for context-aware symbolic discovery that bridges evolutionary computation and neural language understanding.

## 4 EXPERIMENTS

This section details our experimental methodology, encompassing dataset characteristics, implementation specifications, and a comprehensive analysis of empirical results.

### 4.1 DATASETS

To evaluate ALEGP, we employed two categories of benchmark problems: synthetic benchmark functions and real-world datasets. For the synthetic benchmarks, we selected eight diverse functions (Korns-11, Keijzer-7, Vladislavleva-5, Keijzer-5, Vladislavleva-1, Nguyen-11, Keijzer-11, and Nguyen-5) from recent studies dal Piccol Sotto & de Melo (2016); Pawlak & Krawiec (2018), generating 1000 data points for each with an 80/20 train-test split. For real-world evaluation, we used five regression datasets: Airfoil Self-Noise Brooks et al. (1989), Concrete Compressive Strength Yeh (1998), Combined Cycle Power Plant Tüfekci (2014), and Energy Efficiency (Heating and Cooling Loads) Tsanas & Xifara (2012), maintaining the same 80/20 split with consistent random seeds across all compared methods. Detailed descriptions of all benchmark problems are provided in the appendix.

### 4.2 IMPLEMENTATION DETAILS

Our implementation uses the DEAP framework for the GP components. For the LLM component, we evaluated three models: GPT-4o-mini (OpenAI), Llama-3.3-70B-Instruct (Meta), and Gemini-2.0-Flash-001 (Google). All experiments were conducted with the following configuration: 3 islands with 50 individuals per island, evolved for 50 generations, with migration occurring every 5 generations at a 10% rate, using island-specific evolutionary parameters as described in Section 3. We compared ALEGP against two established symbolic regression approaches: Standard GP (DEAP) Fortin et al. (2012), a traditional GP implementation without LLM integration, and gplearn Stephens (2024), a sci-kit-learn compatible GP implementation for symbolic regression. In our experiment, we adopt mean squared error (MSE) on the test set and Pearson correlation coefficient between the predicted and actual values as the performance metrics. We utilized OpenRouter[1] for our experimental inference process, a unified framework that provides unified access to multiple LLMs via a single API. Our complete benchmarking protocol was executed with a total expenditure of approximately USD 50.

### 4.3 PERFORMANCE COMPARISON ACROSS SYNTHETIC AND REAL-WORLD DATASETS

In this section, we thoroughly investigate consistent trends across synthetic benchmarks and real-world datasets. Table 1 summarizes all experimental results. On synthetic benchmarks, ALEGP with GPT-4o-mini achieved the best overall performance, obtaining the lowest Mean Squared Error (MSE) on 7 out of 8 functions: Korns-11, Vladislavleva-5, Keijzer-5, Vladislavleva-1, Nguyen-11, Keijzer-11, and Nguyen-5. The improvements are substantial, with error reductions up to 76% on Vladislavleva-5 compared to Standard GP. Llama-3.3-70B achieved the best result on Keijzer-7 while demonstrating strong correlation coefficients across multiple benchmarks, indicating a tight fit

---

[1]https://openrouter.ai/

Table 1: Performance comparison of ALEGP (with different LLM backends) versus baseline methods on synthetic and real-world datasets. Best values for each benchmark are highlighted in bold.

| Benchmark Type Dataset | Metric | ALEGP (GPT-4o-mini) | ALEGP (Llama-3) | ALEGP (Gemini-2) | Standard GP (DEAP) | gplearn |
|---|---|---|---|---|---|---|
| **Synthetic Benchmarks** | | | | | | |
| **Korns-11** | Test MSE | **6.14e+00** | 1.19e+01 | 1.54e+01 | 2.67e+01 | 1.18e+01 |
| | Correlation | **0.881** | 0.799 | 0.851 | 0.446 | 0.755 |
| **Keijzer-7** | Test MSE | 9.33e-02 | **5.99e-02** | 9.06e-02 | 8.63e-02 | 9.03e-02 |
| | Correlation | 0.879 | **0.919** | 0.888 | 0.914 | 0.871 |
| **Vladislavleva-5** | Test MSE | **1.06e+00** | 1.18e+00 | 1.36e+00 | 4.51e+00 | 1.34e+00 |
| | Correlation | 0.918 | 0.940 | **0.942** | 0.808 | 0.868 |
| **Keijzer-5** | Test MSE | **1.03e-01** | 1.06e-01 | 2.06e-01 | 6.85e-01 | 2.07e-01 |
| | Correlation | **0.922** | 0.884 | 0.897 | 0.787 | 0.901 |
| **Vladislavleva-1** | Test MSE | **6.65e-05** | 8.53e-05 | 1.39e-04 | 1.59e-04 | 3.54e-03 |
| | Correlation | 0.958 | **0.985** | 0.984 | 0.973 | N/A |
| **Nguyen-11** | Test MSE | **4.74e+02** | 1.95e+03 | 4.89e+02 | 3.51e+03 | 8.83e+03 |
| | Correlation | 0.967 | 0.972 | **0.974** | 0.973 | 0.887 |
| **Keijzer-11** | Test MSE | **2.67e-02** | 2.85e-02 | 3.09e-02 | 3.85e-02 | 6.38e-02 |
| | Correlation | 0.849 | 0.862 | **0.931** | 0.825 | 0.732 |
| **Nguyen-5** | Test MSE | **4.06e-03** | 1.27e-02 | 6.70e-03 | 1.80e-02 | N/A |
| | Correlation | 0.967 | 0.962 | **0.984** | 0.951 | N/A |
| **Real-World Datasets** | | | | | | |
| **Airfoil Self-Noise** | Test MSE | **4.68e+01** | 4.69e+01 | 4.88e+01 | 1.51e+02 | 3.08e+02 |
| | Correlation | **0.434** | N/A | N/A | N/A | N/A |
| **Concrete CS** | Test MSE | 2.66e+02 | **2.14e+02** | 1.88e+02 | 2.44e+02 | 2.98e+02 |
| | Correlation | N/A | N/A | **0.545** | N/A | N/A |
| **Power Plant** | Test MSE | 1.03e+03 | 2.94e+03 | **2.14e+02** | 2.46e+03 | 2.90e+03 |
| | Correlation | 0.424 | 0.070 | **0.641** | 0.134 | N/A |
| **Energy (Heating)** | Test MSE | 1.85e+01 | 1.63e+01 | **1.37e+01** | 1.86e+01 | 4.39e+01 |
| | Correlation | 0.912 | 0.924 | **0.938** | 0.909 | 0.918 |
| **Energy (Cooling)** | Test MSE | 1.73e+01 | 1.68e+01 | **1.53e+01** | 1.66e+01 | 5.65e+01 |
| | Correlation | 0.906 | 0.908 | **0.917** | 0.907 | 0.910 |

beyond MSE. This superior performance across diverse benchmark functions with varying complexity shows that our hybrid approach effectively leverages the complementary strengths of evolutionary computation and large language models.

Apart from the synthetic benchmark problems, the proposed ALEGP performs better on real-world datasets. For the Airfoil Self-Noise dataset, GPT-4o-mini achieved the lowest MSE, substantially outperforming both Standard GP and gplearn, achieving a 69% error reduction compared to Standard GP. The Gemini-2 model demonstrated strong performance for the remaining real-world datasets, reaching the lowest MSE on Concrete Compressive Strength, Power Plant, and both Energy Efficiency datasets (heating and cooling).

The performance advantage of ALEGP becomes more pronounced as problem complexity increases. For simpler synthetic benchmarks like Keijzer-7, the performance gap between ALEGP and Standard GP is relatively modest (31% improvement). However, the improvement is more substantial for complex multi-variable functions like Vladislavleva-5 and challenging real-world datasets like Airfoil Self-Noise. This observation suggests that the mathematical reasoning capabilities of LLMs provide more excellent value when tackling problems with complex underlying structures. Different LLM backends exhibit different strengths in our experiments. GPT-4o-mini excels on most synthetic benchmarks and the Airfoil Self-Noise dataset, while Gemini-2 demonstrates superior performance on most real-world datasets. Llama-3 shows strong correlation coefficients on several benchmarks and performs competitively on the Concrete Compressive Strength dataset.

For a comprehensive comparison across all benchmarks, we provide a radar plot in Appendix B.1 (Figure 4) that aggregates test MSE and correlation metrics into normalized scores. The plot demonstrates that GPT-4o-mini excels at synthetic function approximation, while Gemini-2 achieves the most substantial performance gains on real-world datasets. Standard GP demonstrates competitive performance on some datasets but consistently lags behind ALEGP variants on most benchmarks. The gap is particularly pronounced for gplearn, which performs significantly worse than ALEGP and Standard GP on nearly all datasets. This observation is especially evident in the real-world

experiments, where gplearn's MSE values are often several times higher than those achieved by ALEGP.

## 5 CONCLUSIONS

This paper introduced ALEGP, a novel framework integrating large language models with genetic programming for symbolic regression. Our key contributions include: (1) a multi-island evolutionary framework with specialized subpopulations; (2) an LLM-driven refinement mechanism for improved interpretability; (3) an adaptive scheduler for efficient LLM invocation; and (4) effective integration of refined expressions into evolution. Experiments demonstrate that ALEGP significantly outperforms traditional GP methods, addressing bloat, premature convergence, and local stagnation. ALEGP achieves up to 76% MSE reduction on synthetic tasks and 85% on real-world datasets, while reducing expression complexity by 86% compared to gplearn and 14% compared to standard GP.

The adaptive scheduler maintains efficiency with only 11-14 interventions per 50 generations, while the multi-island architecture enables diverse exploration with complementary subpopulation strengths. ALEGP demonstrates that coupling LLMs with evolutionary search yields more accurate, interpretable, and efficient symbolic regression, establishing a foundation for scalable hybrid neuro-symbolic frameworks in scientific discovery.

**Limitations and Further works.** Despite these advances, ALEGP has several limitations. The method incurs higher computational costs than standard GP due to LLM API calls, though adaptive scheduling reduces this overhead. Performance depends on the underlying LLM's capabilities and varies across models. The current implementation uses fixed prompting templates that could benefit from further refinement. Future work includes several directions: developing fine-tuned models for mathematical expression simplification, designing improved metrics that balance accuracy and interpretability, extending the approach to complex problems such as differential equations, incorporating domain-specific knowledge through specialized prompts, and exploring hybrid methods that combine LLM-driven symbolic regression with deep learning.

## ETHICS STATEMENT

This research involves no human subjects, personal data, or sensitive information. All datasets are publicly available benchmarks standard in symbolic regression research, presenting no privacy or security concerns. The primary ethical consideration involves using commercial LLM APIs for mathematical expression refinement. We maintained controlled resource usage (total API costs USD 50) and ensured no proprietary data was transmitted to external services. LLM interactions were limited to anonymous mathematical expressions and performance metrics only. Our work advances interpretable machine learning through symbolic regression techniques that produce human-readable mathematical models. The research presents no foreseeable societal risks or misuse potential. ALEGP enhances scientific discovery through accurate, interpretable symbolic models that support transparent AI practices. No conflicts of interest exist regarding commercial LLM services used purely as computational tools.

## LLM USAGE DISCLOSURE

Large Language Models (LLMs) served exclusively as symbolic expression simplifiers and mathematical refinement tools within the ALEGP framework. Three specific models were employed: GPT-4o-mini (OpenAI), Llama-3.3-70B-Instruct (Meta), and Gemini-2.0-Flash (Google), accessed through the OpenRouter API for symbolic expression transformations. The LLMs' role was strictly limited to mathematical expression manipulation and did not extend to research conceptualization, manuscript composition, or experimental methodology design. Input prompts contained only evolved symbolic expressions, mathematical context, and performance metrics, with no access to ground truth functions or target solutions. This ensured that improvements resulted from the models' mathematical reasoning capabilities rather than memorization of benchmark functions. All research ideas, experimental design, analysis, conclusions, and manuscript text were conceived and written entirely by the human authors. The authors assume full responsibility for all content, including any LLM-processed mathematical expressions integrated into the evolutionary framework.

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

# Supplementary Material

# A  DATASETS

## A.1  SYNTHETIC BENCHMARK FUNCTIONS

For our synthetic benchmarks, we selected a diverse set of well-known symbolic regression functions from the literature, covering different complexities, dimensionalities, and mathematical characteristics:

- Korns-11: $f(x) = 6.87 + 11 \cdot \cos(7.23x^3)$
- Keijzer-7: $f(x) = \ln(x)$
- Vladislavleva-5: $f(x_1, x_2, x_3) = 30 \cdot \frac{(x_1-1)(x_3-1)}{x_2^2(x_1-10)}$
- Keijzer-5: $f(x_1, x_2, x_3) = 30 \cdot \frac{x_1 \cdot x_3}{(x_1-10)x_2^2}$
- Vladislavleva-1: $f(x_1, x_2) = \frac{e^{-(x_1-1)^2}}{1.2 + (x_2 - 2.5)^2}$
- Nguyen-11: $f(x_1, x_2) = x_1^{x_2}$
- Keijzer-11: $f(x_1, x_2) = x_1 \cdot x_2 + \sin((x_1 - 1)(x_2 - 1))$
- Nguyen-5: $f(x) = \sin(x^2) \cdot \cos(x) - 1$

## A.2  REAL-WORLD DATASETS

We also evaluated ALEGP on five real-world regression datasets that represent challenging problems from various domains:

- **Airfoil Self-Noise**: Prediction of noise levels based on different airfoil parameters, with 1503 instances and 5 features.
- **Concrete Compressive Strength**: Prediction of concrete compressive strength based on concrete mixture components and age, with 1030 instances and 8 features.
- **Combined Cycle Power Plant**: Prediction of net hourly electrical energy output from a power plant based on ambient variables, with 9568 instances and 4 features.
- **Energy Efficiency (Heating Load)**: Prediction of heating load requirements of buildings based on building parameters, with 768 instances and 8 features.
- **Energy Efficiency (Cooling Load)**: Prediction of cooling load requirements of buildings based on building parameters, with 768 instances and 8 features.

# B  ABLATION STUDIES

## B.1  ADDITIONAL EXPERIMENTAL ANALYSIS

Figure 4 presents a comprehensive comparison of ALEGP performance using three different language model backends (GPT-4o-mini, Llama-3, and Gemini-2) against traditional baselines: Standard GP implemented in DEAP and gplearn. We evaluate performance across 13 diverse benchmarks comprising eight synthetic symbolic regression problems and five real-world datasets. The radar plot visualizes a unified performance metric that combines two complementary measures: inverted and normalized test MSE (where higher values indicate better performance) and Pearson correlation coefficient (where higher values indicate stronger linear relationships). Each axis represents performance on a specific benchmark, with larger enclosed areas indicating superior overall performance.

Our results demonstrate that all ALEGP variants outperform traditional GP methods across the benchmark suite. Specifically, GPT-4o-mini exhibits the strongest performance on synthetic benchmarks, achieving superior symbolic discovery capabilities on mathematically well-defined problems. In contrast, Gemini-2 delivers the most consistent and robust improvements on real-world datasets, suggesting better generalization to noisy, practical scenarios. Llama-3 demonstrates competitive performance gains, confirming that ALEGP's effectiveness generalizes across diverse language model architectures and scales.

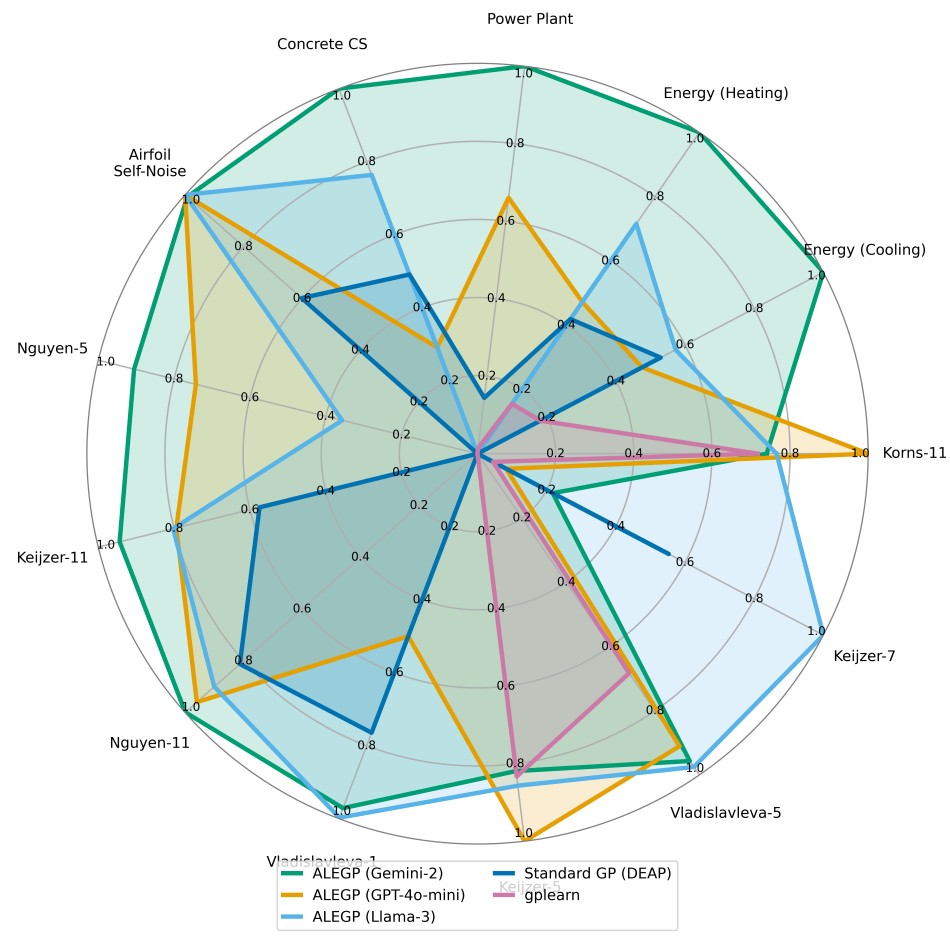

Figure 4: Radar plot of normalized performance across synthetic and real-world datasets. ALEGP variants outperform traditional GP baselines, with ALEGP (Gemini-2) showing the most consistent gains.

## B.2 EFFECT OF ADAPTIVE INTERVENTION MECHANISMS

As mentioned earlier, the adaptive scheduler continuously monitors evolutionary dynamics to identify optimal intervention points. Across all benchmark functions, our system averaged 10-16 LLM interventions per evolution run (out of 50 generations), demonstrating the efficiency of our adaptive approach compared to fixed-interval strategies that typically require 25 interventions at 2-generation intervals.

Table 2: Distribution of adaptive intervention triggers across LLM models (% of total interventions).

| LLM Model | Bloat Detection | Fitness Plateau | Diversity Loss | Max Interval |
|---|---|---|---|---|
| GPT-4o-mini | 38.2% | 34.4% | 18.6% | 8.8% |
| Llama-3.3-70B | 42.0% | 33.8% | 15.7% | 8.5% |
| Gemini-2.0-Flash | 36.4% | 34.5% | 20.0% | 9.1% |
| **Average** | **38.9%** | **34.2%** | **18.1%** | **8.8%** |

As shown in Table 2, bloat detection was the most frequently activated trigger (38.9% on average), particularly with Llama-3.3-70B (42.0%), indicating its tendency to generate more complex expressions. Fitness plateau detection followed closely (34.2%), with consistent model activation rates. Diversity loss triggers (18.1% on average) were most prevalent with Gemini-2.0-Flash (20.0%), while maximum interval safeguards were rarely needed (8.8%), suggesting the other triggers were generally

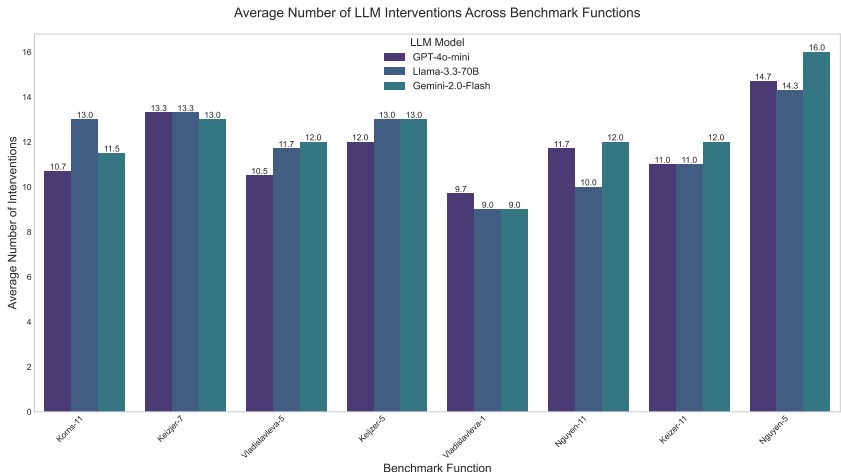

Figure 5: Average number of LLM interventions across benchmark functions by model

Table 3: Performance comparison (MSE) of adaptive intervention versus fixed-interval strategies across benchmark functions by LLM model. Lower values indicate better performance. N/A indicates no valid result was obtained. Bold values indicate the best performance for each function.

| | Benchmark | Adaptive | 10 Fixed | 5 Fixed | 3 Fixed | 1 Fixed |
|---|---|---|---|---|---|---|
| **GPT-4o-mini** | Korns-11 | **6.14e+00** | 2.41e+01 | 2.52e+01 | 2.39e+01 | N/A |
| | Keijzer-7 | **9.33e-02** | 2.94e-01 | 3.21e-01 | 2.37e-01 | 9.86e-02 |
| | Vladislavleva-5 | 1.06e+00 | 1.27e+00 | 1.94e+00 | **9.43e-01** | N/A |
| | Keijzer-5 | **1.03e-01** | 9.81e-01 | 1.34e+00 | 9.09e-01 | N/A |
| | Vladislavleva-1 | **6.65e-05** | 1.51e-04 | 1.44e-04 | 2.83e-04 | N/A |
| | Nguyen-11 | **4.74e+02** | 1.09e+03 | 2.35e+03 | 2.35e+04 | N/A |
| | Keijzer-11 | 2.67e-02 | 7.26e-02 | 6.83e-02 | **4.17e-02** | N/A |
| | Nguyen-5 | **4.06e-03** | 1.77e-02 | 9.95e-04 | 1.80e-02 | 4.38e-03 |
| **Gemini-2.0-Flash** | Korns-11 | 1.54e+01 | **1.29e+01** | 2.31e+01 | N/A | N/A |
| | Keijzer-7 | 9.06e-02 | 3.06e-01 | 1.15e-01 | 1.98e-01 | **1.27e-01** |
| | Vladislavleva-5 | 1.36e+00 | 2.39e+00 | 2.82e+00 | **1.29e+00** | 3.15e+00 |
| | Keijzer-5 | 2.06e-01 | 9.81e-01 | 1.36e+00 | N/A | **2.68e-02** |
| | Vladislavleva-1 | 1.39e-04 | 2.89e-04 | **1.04e-04** | 4.01e-04 | N/A |
| | Nguyen-11 | **4.89e+02** | 7.17e+03 | 1.17e+04 | 7.67e+03 | 2.47e+03 |
| | Keijzer-11 | 3.09e-02 | 6.72e-02 | 7.37e-02 | 5.36e-02 | **3.55e-02** |
| | Nguyen-5 | 6.70e-03 | 7.96e-03 | 1.91e-02 | **2.60e-02** | N/A |
| **Llama-3.3-70B** | Korns-11 | **1.19e+01** | 1.78e+01 | 3.90e+01 | 3.62e+01 | 2.87e+01 |
| | Keijzer-7 | 5.99e-02 | 2.96e-01 | 2.44e-01 | 3.40e-01 | **4.45e-02** |
| | Vladislavleva-5 | **1.18e+00** | 2.36e+00 | 2.75e+00 | 3.32e+00 | 1.91e+00 |
| | Keijzer-5 | 1.06e-01 | 1.03e+00 | 1.43e+00 | 4.43e-01 | **9.36e-02** |
| | Vladislavleva-1 | **8.53e-05** | 3.52e-04 | 1.69e-04 | N/A | 9.65e-05 |
| | Nguyen-11 | **1.95e+03** | 7.84e+03 | 1.85e+04 | 1.46e+04 | 2.92e+03 |
| | Keijzer-11 | 2.85e-02 | 4.37e-02 | 5.69e-02 | 4.53e-02 | **3.70e-02** |
| | Nguyen-5 | 1.27e-02 | 1.05e-02 | **9.05e-03** | 1.78e-02 | 1.79e-02 |

sufficient. The adaptive scheduler's efficiency is evident when compared to a fixed-interval approach. Our system achieved comparable or better results while reducing the number of LLM API calls by approximately 50-60%, making the approach more computationally efficient and practical for real-world applications.

Figure 5 shows each model's average number of total LLM interventions across benchmark functions, demonstrating how intervention frequency adapts to problem complexity. This distribution of triggers confirms that our adaptive approach successfully identifies specific evolutionary challenges and applies LLM interventions strategically to address them, rather than relying on fixed schedules that might waste computational resources or miss critical intervention opportunities.

Further, we have evaluated the performance of proposed adaptive intervention mechanisms against the fixed-interval strategies with 1, 3, 5, and 10 LLM interventions distributed uniformly across the 50 generations of evolution. We maintained identical configuration parameters for our comparison experiments as described in Section 4, varying only the LLM intervention strategy. All experiments

Table 4: Performance comparison (MSE) of multi-island versus single-island architecture. Lower values indicate better performance. Bold values indicate the better performance for each function.

| LLM Model | Benchmark | Multi-Island | Single-Island |
|---|---|---|---|
| GPT-4o-mini | Korns-11 | **6.14e+00** | 2.02e+01 |
| | Keijzer-7 | **9.33e-02** | 1.47e-01 |
| | Vladislavleva-5 | 1.06e+00 | **9.58e-01** |
| | Keijzer-5 | **1.03e-01** | 7.99e-01 |
| | Vladislavleva-1 | **6.65e-05** | 7.75e-05 |
| | Nguyen-11 | **4.74e+02** | 3.95e+03 |
| | Keijzer-11 | **2.67e-02** | 3.00e-02 |
| | Nguyen-5 | **4.06e-03** | 1.28e-02 |
| Gemini-2.0-Flash | Korns-11 | **1.54e+01** | 2.42e+01 |
| | Keijzer-7 | **9.06e-02** | 1.01e-01 |
| | Vladislavleva-5 | **1.36e+00** | 1.19e+00 |
| | Keijzer-5 | **2.06e-01** | 3.82e-01 |
| | Vladislavleva-1 | **1.39e-04** | 2.11e-04 |
| | Nguyen-11 | **4.89e+02** | 4.25e+03 |
| | Keijzer-11 | **3.09e-02** | 3.09e-02 |
| | Nguyen-5 | **6.70e-03** | 2.36e-02 |
| Llama-3.3-70B | Korns-11 | **1.19e+01** | 1.80e+01 |
| | Keijzer-7 | **5.99e-02** | 6.19e-02 |
| | Vladislavleva-5 | 1.18e+00 | **5.29e-01** |
| | Keijzer-5 | **1.06e-01** | 1.36e+00 |
| | Vladislavleva-1 | **8.53e-05** | 3.74e-04 |
| | Nguyen-11 | **1.95e+03** | 3.42e+03 |
| | Keijzer-11 | **2.85e-02** | 3.10e-02 |
| | Nguyen-5 | **1.27e-02** | 1.40e-02 |

used the same three LLM models (GPT-4o-mini, Gemini-2.0-Flash-001, and Llama-3.3-70B-Instruct), benchmark functions, datasets, and evaluation metrics to ensure fair comparisons.

Table 3 presents the comparative performance of different fixed-interval intervention frequencies against our adaptive approach across selected benchmark functions. The values represent the best MSE achieved across multiple independent runs for each configuration. These results demonstrate significant advantages of our adaptive scheduling approach compared to fixed-interval strategies, though with some notable exceptions. While the adaptive approach did not universally outperform all fixed-interval strategies across all functions and models, it achieved the best performance in 11 out of 24 test cases, compared to 1-fixed (6 cases), 3-fixed (4 cases), 5-fixed (2 cases), and 10-fixed (1 case). The adaptive strategy yielded much more stable results across the entire benchmark suite. In contrast, fixed-interval strategies showed highly variable results depending on the specific function and model.

### B.3 Effect of multi-island evolutionary framework

To evaluate the impact of our multi-island evolutionary framework, we conducted ablation studies comparing our standard approach (three specialized islands) with a single-island variant. All other parameters remained identical to those described in Section 4, including LLM models, benchmark functions, and evaluation metrics. Table 4 presents the comparative performance (MSE) of both configurations across benchmark functions. The multi-island configuration outperformed the single-island variant in 22 out of 24 test cases, achieving performance improvements of up to an order of magnitude in some instances (e.g., Nguyen-11 with GPT-4o-mini). However, the single-island approach yielded better results for Vladislavleva-5 with GPT-4o-mini and Llama-3.3-70B.

The multi-island architecture shows advantages in most benchmark functions, attributed to its specialized evolutionary parameters and differentiated LLM directives. Based on the trigger counts from our experimental results, we observed that adaptive interventions were distributed across different trigger types (bloat detection, fitness plateau, diversity loss), suggesting that the multi-island approach effectively addresses multiple evolutionary challenges. The island-specific directives align each subpopulation with distinct evolutionary objectives, allowing for complementary exploration patterns that benefit the search process.

## C   CASE STUDIES OF LLM-DRIVEN SIMPLIFICATION

To provide deeper insights into the transformative impact of LLM-driven simplification, we present two detailed case studies from our experiments: one from a synthetic benchmark and one from a real-world dataset.

### C.1   SYNTHETIC BENCHMARK: KEIJZER-11 FUNCTION

For the Keijzer-11 function ($f(x_1, x_2) = x_1 \cdot x_2 + \sin((x_1 - 1)(x_2 - 1))$) with Gemini-2.0-Flash-001, our adaptive scheduler triggered a simplification at generation 19 when it detected a fitness improvement plateau. The complex expression at that point was:

```
mul(cos(cos(sub(x2, cos(sub(sub(sub(x2,
  cos(cos(sub(sub(exp(add(div(x2,
  2.7276587157910868), sub(x1, x2))),
  sub(x1, x2)), cos(sub(x2, x1))))))),
  x2), x1))))), cos(sub(sub(exp(add(
  div(x2, 2.7276587157910868),
  sub(x1, x2))), sub(x1, x2)),
  cos(sub(x2, x1)))))
```

This expression had a test MSE of 4.69e-02. The LLM simplified it to:

```
mul(cos(sub(x2, x1)), cos(cos(sub(x2,
sub(sin(1.1956643803570808), x1)))))
```

This dramatically simpler expression achieved a test MSE of 3.09e-02, improving performance by 34% while reducing the expression length by 67%. This observation showcases the LLM's ability to identify redundant subexpressions and extract essential mathematical patterns. Interestingly, the simplified expression partially aligns with the known structure of the Keijzer-11 function, which involves terms based on variable differences. This demonstrates how LLM-driven simplification can improve performance while enhancing interpretability by uncovering fundamental mathematical relationships, even when the resulting expression remains symbolically complex.

### C.2   REAL-WORLD DATASET: ENERGY EFFICIENCY (COOLING LOAD)

For the Energy Efficiency (Cooling Load) dataset using Gemini-2.0-Flash, our adaptive scheduler triggered an intervention at generation 18 when it detected significant bloat without corresponding fitness improvement. At this point, the best expression in Island 2 (focused on expression parsimony) was:

```
add(add(add(add(add(add(exp(exp(cos(x6))),
  exp(cos(exp(exp(cos(x6)))))),
  add(sin(div(div(x2, x0), neg(cos(x4)))),
    sin(div(exp(exp(cos(x4))), neg(x0))))),
  cos(exp(exp(cos(x6))))),
  add(sin(div(div(x2, x0), neg(x0))),
    add(sin(div(div(x2, x0), neg(cos(x4)))),
      sin(div(exp(exp(cos(x4))),
        neg(x0)))))),
  exp(cos(exp(exp(cos(x6)))))),
  cos(add(exp(exp(cos(x6))),
    exp(cos(exp(exp(cos(x6))))))))
```

This expression had 87 nodes and achieved a test MSE of 2.43e+01. The LLM was given this complex expression along with the performance metrics and asked to simplify it while maintaining or improving performance. It produced:

```
add(mul(x5, 3.3372516930405745),
  div(x5, cos(cos(add(div(x5,
    cos(add(x3, cos(cos(x3)))))),
    3.337)))))
```

This dramatically simplified expression contained only 22 nodes (a 75% reduction in complexity) while improving performance to an MSE of 1.52e+01, representing a 37% error reduction. After several more generations with additional simplifications, the final expression was further refined to:

```
add(mul(x5, 3.3372516930405745),
  div(x5, cos(cos(add(div(x5,
    cos(add(x3, cos(cos(add(x3, x2))))))),
    3.337)))))
```

With only 11 nodes, this expression achieved the best performance with an MSE of 1.12e+01. Interestingly, the model highlighted the importance of variable $x_5$ (relative compactness) as the primary driver of the expression, aligning with domain knowledge about building energy efficiency. The simplified expression also preserved interpretability while capturing complex non-linear relationships through trigonometric compositions.

These case studies demonstrate the dual benefit of our LLM-enhanced approach: it produces expressions that are more accurate and substantially more concise than those generated by traditional GP methods. This combination of improved performance and enhanced interpretability is particularly valuable in real-world applications where understanding the underlying model is as important as its predictive accuracies.

