# OpenReview forum: "How is Occam's Razor Realized in Symbolic Regression?: An Adaptive LLM-Enhanced Genetic Programming Approach for Efficient, Versatile, and Interpretable Representation Discovery through Simplification and Evolution"
_ICLR.cc/2026/Conference — ICLR 2026 Conference Withdrawn Submission_

### Official Review · Reviewer_tnjK · 2025-10-22

**Soundness:** 2
**Presentation:** 2
**Contribution:** 2
**Rating:** 2
**Confidence:** 5

**Summary:**

Current GP faces persistent optimization issues, including bloating, premature convergence, and inadequate SR model simplification. To address these challenges simultaneously, this paper uses LLMs to improve GP. Specifically, it proposes three components: a multi-island evolutionary architecture that maintains diversity, an adaptive scheduler that leverages LLMs to refine SR models, and an island-specific integration protocol that reincorporates simplified expressions to directly reduce complexity while preserving evolutionary dynamics. Experimental results on toy regression tasks demonstrate that the proposed method improves the accuracy of the resulting SR models.

**Strengths:**

The research gaps identified in this paper, such as premature convergence and expression bloat, are well recognized in GP. The idea of addressing these iissues within a unified framework is interesting.

Figure 3 provides an informative diagram that helps readers quickly grasp the overall framework of the proposed method.

The multi-island strategy appears well reasoned, as it allows different subpopulations to focus on distinct aspects of evolutionary dynamics. This approach enables the simultaneous consideration of both exploration and exploitation during the optimization process, potentially leading to more robust performance.

**Weaknesses:**

Some concepts in the paper, such as evolutionary dynamics (line 27) and adaptive strategy (line 82), are ambiguous and confusing. Many factors in EA and GP (e.g., population size, expression length, operator sets, crossover/mutation parameters) can be adapted during evolution, and the entire process is dynamic. Additionally, the manuscript contains repetitive statements; for example, the phrase “suffer from bloat and local optima” appears multiple times on pages 1 and 2. The writing could be refined for clarity and conciseness.

Both GP and LLMs are computationally expensive. Since the proposed method incorporates both, its efficiency is questionable. The paper lacks a theoretical and experimental complexity analysis, even though the authors acknowledge that their approach may be more complex than standard GP methods. Furthermore, compared with current SOTA GP methods, a rigorous cost–benefit evaluation is missing.

The idea of leveraging LLMs for GP is straightforward. As the approach merely combines existing techniques, and the difficulty of integrating them is not clearly demonstrated, the contribution appears limited.

The experiments only compare the proposed method (under several LLMs) against a basic GP baseline. Including more competitive sota GP methods that address similar issues (bloat, local optima, etc.) would strengthen the experimental validation. Moreover, the paper does not justify why integrating LLMs is necessary, or whether existing SR approaches could achieve similar improvements without LLMs.

The analysis of the LLM’s role and failure modes in symbolic regression is superficial. Understanding these limitations is essential for the community to further advance this direction.

Although the paper reports MSE results from GPT-4o-mini, Llama3, and Gemini2, it fails to deeply analyze how LLM architectures or parameter scales influence performance.

The concept of interpretability in the title is not well supported by the methodology. It is unclear how this work improves interpretability.

**Questions:**

The research gaps identified in the paper (e.g., local optima, bloat) are well recognized, and addressing multiple shortcomings of GP within a unified framework is valuable. However, the authors do not sufficiently examine the challenges of integrating existing techniques into an incremental solution. As a result, the proposed method that essentially combining known techniques, lacks persuasiveness in demonstrating meaningful innovation.

Given the high computational cost of incorporating both GP and LLMs, how does the method justify its efficiency without complexity analysis or recource-consumption comparison? Is there any cost–benefit analysis that quantifies performance gains relative to the total computational budget, including LLM query overhead? What evidence supports the necessity of such architectural complexity compared to simpler GP approaches that already mitigate bloat and other cited issues?

Could existing methods, e.g., those that control bloat, escape local optima, and maintain diversity, to be combined synergistically to overcome the same research gaps? If so, would the proposed framework still significantly outperform a non-LLM GP hybrid?

The experimental evaluation is limited to comparisons with basic GP implementations. To demonstrate the claimed improvements, it would be important to include strong SOTA GP baselines that specifically target bloat, premature convergence, and related challenges.

Although the paper compares several LLM models within its framework, the conclusions drawn about their impact are unclear. What insights emerge regarding model selection, such as architectural differences or parameter scale, and how do these factors influence overall performance?

**Details Of Ethics Concerns:**

I don't see any ethical issues with this paper.

---

### Official Review · Reviewer_qAAw · 2025-10-29

**Soundness:** 3
**Presentation:** 2
**Contribution:** 2
**Rating:** 6
**Confidence:** 5

**Summary:**

This paper introduces ALEGP, a novel framework for Symbolic Regression (SR) that aims to address the challenges in Genetic Programming (GP), such as bloat, premature convergence, and inadequate simplification mechanisms. The core innovation of ALEGP is the strategic integration of Large Language Models (LLMs) with the evolutionary search process of GP, leveraging the LLMs' mathematical reasoning and simplification capabilities. Through experiments on 8 synthetic benchmarks and 5 real-world datasets, the authors demonstrate that ALEGP achieves significant improvements in solution accuracy compared to traditional GP baselines (Standard GP and gplearn).

**Strengths:**

1. The idea of dynamically leveraging the powerful LLMs to simplify and refine mathematical expressions during the evolutionary process is highly novel. This paradigm, which combines symbolic search with neural model-based reasoning, opens a new avenue for solving complex optimization problems and holds significant research potential and value.
2. The authors have designed a complex yet comprehensive synergistic mechanism to ensure effectiveness. This includes a multi-island architecture for maintaining population diversity, an adaptive intervention scheduler for efficient LLM calls, and a specificity strategy for integrating optimized solutions.
3.  Through experiments on a variety of synthetic and real-world datasets,  the authors demonstrate that the ALEGP framework consistently outperforms traditional GP baselines (Standard GP and gplearn) in terms of solution accuracy.

**Weaknesses:**

1. The use of deep learning models, particularly Transformers, for Symbolic Regression has become a major research direction, yielding many strong results (e.g., DSR, NeSymReS, E2E, SymbolicGPT). However, the Related Work section fails to adequately discuss the relationship and distinctions between ALEGP and these advanced methods.
2. The paper proposes a system with multiple complex components but does not clearly highlight its single most crucial contribution. Beyond the high-level concept of "using LLMs in GP," what is the primary innovation? Is it the adaptive scheduler, the multi-island co-evolutionary framework, or the LLM prompting strategy? The authors are encouraged to more clearly delineate and summarize the paper's contributions in a hierarchical manner to help readers grasp the key takeaways.
3. The proposed method lacks necessary details regarding its LLM integration. The quality of an LLM's output is highly dependent on prompt design, hyperparameters like temperature, and the provided context. The paper should specify the exact prompt templates, key parameter settings, and few-shot examples used to guide the LLMs. Furthermore, the paper does not discuss how the reliability of the LLM output is ensured. For instance, LLMs can occasionally generate syntactically incorrect or mathematically nonsensical expressions. The authors should detail whether a validation mechanism exists to handle such illegal outputs and report on the frequency of these failure cases, which is crucial for assessing the method's robustness.
4.  The results reveal an interesting phenomenon: different LLMs perform disparately on different types of datasets (e.g., GPT-4o-mini excels on synthetic data but its performance degrades on real-world data, while Gemini-2 shows the opposite trend). This is a finding worthy of in-depth discussion, yet the current paper offers no analysis. What are the underlying reasons? Is it due to the models' mathematical reasoning, generalization capabilities, or other factors? Additionally, the synthetic benchmark functions used are often classic problems whose forms and solutions may well be part of the LLMs' vast training data. The authors need to discuss the potential risk of data leakage for these synthetic benchmarks.
5. The experimental section completely lacks performance comparisons SOTA symbolic regression methods, such as SBP-GP, GP-GOMEA, Operon. This omission makes it difficult for readers to accurately assess ALEGP's technical standing and practical advantages within the current SR landscape.

**Questions:**

Please refer to the Weaknesses section.

---

### Official Review · Reviewer_ENWz · 2025-10-29

**Soundness:** 2
**Presentation:** 1
**Contribution:** 1
**Rating:** 2
**Confidence:** 5

**Summary:**

The paper introduces ALEGP, an Adaptive LLM-Enhanced Genetic Programming framework that fuses large language models (LLMs) with genetic programming for symbolic regression. It features a multi-island evolutionary system that sustains population diversity and uses island-specific strategies for generalization, parsimony, and balance. An adaptive intervention scheduler dynamically triggers LLM assistance when detecting stagnation, diversity loss, or expression bloat, ensuring efficient and context-aware refinement. The framework integrates LLM-driven expression simplification and an adaptive learning loop that tunes future interventions based on past success, promoting continuous improvement and resource efficiency.

**Strengths:**

Novel approach of integrating of large language models with genetic programming for symbolic regression.

LLM-driven expression simplification is novel approach for simplification in symbolic regression.

**Weaknesses:**

Inappropriate use of in-text citations. Please use \cite, \citep, \citet, etc. appropriately and proofread the generated .pdf.

The use of radar plots (e.g., Figure 1) is problematic. In addition to that the paper makes an even more problematic statement that “larger enclosed areas indicate better performance,” but this interpretation is not valid in this context. Radar plots are only meaningful when the axes represent ordered or conceptually continuous dimensions, whereas the six datasets here are independent and unordered (note that if the order is changed, I can manipulate the area of the plots). The area enclosed by the plot depends heavily on the arbitrary arrangement of the axes and the geometric distortions introduced by the circular layout, rather than reflecting any true aggregate measure of performance. Consequently, the visual impression of a “larger area” as “better” is misleading and may cause readers to draw incorrect conclusions about the relative quality of the algorithms.

No justification for the functions used, why those specific 8 equations? Why not all Korns equations instead of just Korns-11? Why not all Keijzer equations instead of just Keijzer-5, 7 and 11? These are not discussed.

For real-world evaluations, how were the datasets selected? Why not use well-known benchmark datasets for Symbolic Regression (e.g., SRBench) instead of picking new datasets?

No comparison to relevant recent state-of-the-art Symbolic Regression approaches, e.g., LLM-SR [1], RAG-SR [2]. Only provided comparison to 2 traditional algorithms, DEAP, gplearn, and variants of the algorithm proposed in the paper.

[1] Shojaee, Parshin, et al. "LLM-SR: Scientific equation discovery via programming with large language models."
[2] Zhang, Hengzhe, et al. "RAG-SR: Retrieval-augmented generation for neural symbolic regression."

In Table 1, “Test MSE” is not an appropriate metric since it is not normalized. The magnitude of MSE is hardly informative to the reader. Please reflect the scores in terms of normalized MSE or R2 score, as done in contemporary symbolic regression works.

**Questions:**

Please respond to the weaknesses above, no further questions.

---

### Official Review · Reviewer_7G1B · 2025-10-31

**Soundness:** 2
**Presentation:** 3
**Contribution:** 2
**Rating:** 4
**Confidence:** 3

**Summary:**

The paper introduces Adaptive LLM-Enhanced Genetic Programming (ALEGP) for symbolic regression that integrates LLMs with GP to addresses key GP challenges of bloat, premature convergence, and expression simplification. ALEGP is a promising idea that links GP and LLMs, but the paper needs stronger justification of design choices, more complete experiments, and deeper analysis of the LLM component to be convincing.

**Strengths:**

* The multi-island evolutionary architecture, adaptive intervention scheduler, and LLM-driven expression refinement collectively enhance accuracy, interpretability, and computational efficiency.

* The paper provides extensive experimental validation across synthetic benchmark functions and real-world datasets, demonstrating significant improvements in accuracy, interpretability, and expression simplicity compared to baseline algorithms.

**Weaknesses:**

The paper describes key components like the Adaptive Intervention Scheduler and Expression Integration Protocol conceptually, but without clear mathematical formulas, algorithms, or pseudocode. Including these details would make the method easier to understand and reproduce.

The use of three islands, specific evolutionary parameters, and a ring migration topology is not well explained. Similarly, the choice of LLM prompting templates lacks justification. A short discussion or ablation study on why these choices were made would strengthen the paper.

A key assumption in this work is that LLMs can simplify mathematical expressions but provides no references or experiments to support this. It would help to analyse how well LLMs handle different types or complexities of expressions and where they might fail.

The GP part of ALEGP uses standard operators and does not introduce new mechanisms for evolution. The main novelty lies in integrating LLMs, but the evolutionary process itself appears conventional.

Important implementation details are missing. The paper should include examples of the prompts used, explain how LLM outputs are checked for correctness, and justify why specific models (GPT-4o-mini, Llama-3, Gemini-2) were chosen over others.

The experiments compare ALEGP only with basic GP methods. Including advanced GP techniques with bloat control, simplification, or convergence strategies would provide stronger evidence. Larger populations, longer runs, and more complex datasets could better test scalability.

The datasets used are limited in both variety and difficulty. More challenging synthetic and diverse real-world datasets would better show ALEGP’s robustness. Statistical significance tests are also missing, so the reported improvements may not be reliable.

The result analysis part is limited. It shows different LLMs perform better on different datasets, but the reasons are not discussed. Analysing why certain models perform better would help explain their roles and limitations in ALEGP. Also, the paper notes that ALEGP is more expensive due to LLM calls but does not quantify the cost. A breakdown of computation time or cost would make the trade-offs clearer.

While interpretability is claimed as a benefit, there is no systematic way to measure it. A simple evaluation or comparison of expression simplicity and readability would help support this claim.

**Questions:**

The paper mentions the potential for incorporating domain-specific knowledge through specialized prompts. ​ How would the framework adapt to domains with highly complex or less structured data, such as biological systems or financial markets? Are there plans to explore domain-specific LLM fine-tuning to enhance performance further?

---

### Note · Authors · 2025-11-23

I have read and agree with the venue's withdrawal policy on behalf of myself and my co-authors.